# OpenReview forum: "The Impact of Inference Acceleration Strategies on Bias of Large Language Models"
_NeurIPS.cc/2024/Workshop/SafeGenAi — SafeGenAi Poster_

### Official Review · Reviewer_L81z · 2024-10-08

**Rating:** 6
**Confidence:** 4

**Review:**

**Strength**

The paper delves into the impact of inference acceleration techniques on demographic biases within LLM outputs. It offers a comprehensive analysis across multiple metrics and demonstrates that the application of acceleration methods can significantly affect bias, emphasizing the need for careful, case-by-case evaluation of models after such optimizations.

**Weakness**

1. The paper provides limited discussion on the characteristics of various state-of-the-art LLMs and the specific inference acceleration strategies employed, which are crucial for understanding their performance across the selected bias evaluation metrics. Furthermore, sampling the output only five times and reporting the average bias for non-greedy decoding may not be sufficient to fully assess the robustness of the model. A more extensive evaluation, with a larger sample size, would offer greater confidence in the results and the model’s stability across different scenarios.
2. The paper lacks a systematic qualitative and quantitative analysis of the underlying reasons for the varying bias tendencies observed with different inference acceleration strategies in LLMs. While the authors present results based on evaluation metrics, they stop short of offering deeper insights into the causes of these behaviors. Instead, they conclude that the impact of individual acceleration strategies is inconsistent across models and can be unpredictable. This lack of a thorough explanation weakens the overall conclusion and makes the findings less convincing, as it does not provide a clear understanding of why such variations occur.

**Question**
1. In Table 2, Table B.1, Table B.2 and Table C.1, authors mention "A value of ND means there wasnot enough data for this combination", however, it never shows up in the tables. Instead, there are hyphens used. I believe notations should be consistent here.
2. In Figure B.1, the world map shows the comparison of similarity to the opinions of respondents from prompted countries. Authors could mention that how many countries were included in this analysis, and introduce the similarity metric used.
3. In Figure B.1 (a) caption, there's a typo in "base model to to the opinions", which double typed "to".

---

### Official Review · Reviewer_iqFN · 2024-10-09
**This paper provides a valuable evaluation of the impact of inference acceleration strategies on demographic bias in large language models, highlighting the importance of this perspective for future research and development.**

**Rating:** 7
**Confidence:** 4

**Review:**

Thank you for submitting this notable paper. This work explores the demographic bias introduced by the application of inference acceleration strategies to large language models.

It is encouraging to see a comprehensive comparison across different inference acceleration strategies against different bias metrics, which clearly shows that these acceleration techniques may cause a significant increase in some bias metrics. The table summarizing the results is informative.

I would recommend this paper to be accepted as it provides a good new angle to evaluate the inference acceleration strategies and can be beneficial to future research and the development of new strategies.

---

### Official Review · Reviewer_38o8 · 2024-10-10

**Rating:** 6
**Confidence:** 4

**Review:**

**Summary**

The paper explores how efficient inference techniques (such as quantization) affects bias on LLMs. Through a comprehensive empirical evaluation, it measures biases from several different aspects across multiple LLMs and benchmarks. The work has practical implications for balancing efficiency and ethical considerations in LLM deployment. However, the intuition of why these discoveries occur is not well discussed.


**Strengths**
- This paper studies an important problem which is potentially overlooked by the community, which is whether efficient inference techniques make LLMs more prone to biases.

- The paper is well-written and easy to follow.

- The evaluation is comprehensive, including various efficient inference techniques and dimensions of bias.


**Weaknesses**
- The evaluated LLMs are a bit limited to LLaMA and Mistral typed models. Replacing LLaMA2 to a different style of model would make the conclusions even more general.

- While the paper presents large amount of evaluation results on different benchmarks,  it does not explore why these effects occur. Additional analysis such as interpretability analysis, change of internal model representations, could enhance the understanding of these biases.

- There is no solution proposed. Adding a small paragraph discussing potential solutions or directions, preferably based on the findings of the analysis, would further enhance and improve this work.